# Interventions by Rehabilitation Nurse Specialists in the Training of Informal Carers of Older People at Home with Chronic Diseases: A Scoping Review

**DOI:** 10.3390/ijerph22070971

**Published:** 2025-06-20

**Authors:** Ana Rita Bento, Ana Rita Duque, Nelson Gonçalves, Paulo Vaz, Susana Calção, Vanessa Benedito, Rogério Ferreira, César Fonseca, Celso Silva

**Affiliations:** 1São João de Deus School of Nursing, University of Évora, 7000-811 Évora, Portugal; artbento@ulsac.min-saude.pt (A.R.B.); aduque@ulsac.min-saude.pt (A.R.D.); nmgoncalves@ulsalg.min-saude.pt (N.G.); pavaz@ulsac.min-saude.pt (P.V.); scaramelo@ulsac.min-saude.pt (S.C.); vanessa.benedito@ulsba.min-saude.pt (V.B.); cfonseca@uevora.pt (C.F.); 2Évora Espírito Santo Hospital, Alentejo Central Local Health Unit, 7000-811 Évora, Portugal; 3Faro Hospital, Algarve Local Health Unit, 8000-386 Faro, Portugal; 4José Joaquim Fernandes Hospital, Baixo Alentejo Local Health Unit, 7801-849 Beja, Portugal; 5Comprehensive Health Research Centre—CHRC-UÉ, University of Évora, 7000-811 Évora, Portugal; ferrinho.ferreira@ipbeja.pt; 6School of Health, Polytechnic Institute of Beja, 7800-295 Beja, Portugal

**Keywords:** informal caregiver, rehabilitation nursing, empowerment, older adult, home care, quality of life

## Abstract

Background: The aging population is increasing, leading to a greater need for home care for older adults, often provided by informal caregivers (ICs). These caregivers face numerous challenges, requiring adequate training and support. Objectives: This study aimed to map the main interventions performed by the Rehabilitation Nursing Specialist in empowering ICs of older adults at home. Methods: A scoping review was conducted following the Joanna Briggs Institute methodology. The search included seven articles published between 2019 and 2024, in Portuguese, English, and Spanish, available in the PubMed e CINHAL Ultimate databases. The descriptors used were (Rehabilitation Nursing) AND (Informal Caregivers OR Caregivers) AND (Elderly OR Aged) AND (mentoring OR Training. Results: The RNS interventions focused on training caregivers in technical skills (e.g., positioning, transfers, hygiene care, feeding, medication administration), preventing caregiver burden, managing behavioral and psychological symptoms of dementia, promoting self-care, and emotional support. Educational programs and the use of technologies (telehealth) were identified as effective strategies. Conclusions: RNS interventions are crucial for enhancing the skills and well-being of ICs, improving the quality of care provided to older adults at home, and reducing caregiver burden. Person-centered care, continuous support, and recognizing the caregiver’s role are fundamental aspects of these interventions.

## 1. Introduction

Population aging is a global trend, with the expectation that by 2030 around 12% of the world’s population will be made up of people aged 60 or over, which represents approximately 1.4 billion individuals [1].

All over the world, rehabilitation nurses play a crucial role in training informal carers, which significantly improves the quality of life of the elderly. By focusing on education and support, these nurses empower carers to effectively manage the complexities of elderly care. This training encompasses various aspects, including self-care strategies, safety measures and emotional support, which are essential for both the carer and the elderly person.

Rehabilitation nurses can train informal carers, providing education on self-care strategies, enhancing physical capacity, cognitive function and health literacy, ultimately improving the quality of life of the elderly person and ensuring continuity of care throughout the rehabilitation process [2].

Nurses train informal carers, identifying and eliminating architectural barriers, ensuring a safe environment and monitoring the condition of the elderly, ultimately enhancing carers’ skills and improving the autonomy and quality of life of the elderly in general [3].

In Portugal, according to the 2021 Census, released by the National Statistics Institute in 2022, the aging of the population has worsened, with a significant increase in the number of elderly people (23.4% of the resident population in Portugal are aged 65 and over). In 2021, there were 182 elderly people for every 100 young people [4].

When discussing population aging, it is important to recognize the diversity of health experiences. Although some individuals may face health challenges, including comorbidities or changes in functional capacities, aging is not inherently synonymous with dependency or uniform decline. Health needs can vary widely and are influenced by individual circumstances, accessibility to preventive care and social support systems [5].

The increase in the number of older people requires more resources and health services, which can overburden the health system, requiring a constant search for solutions to integrate them. Many of these solutions involve institutionalizing these people; however, it is estimated that there are around 800,000 ICs in Portugal, with approximately 8% of the population providing care for family members and those close to them without receiving any kind of income [6].

In September 2019, the Informal Caregiver Statute was approved, with the aim of regulating the rights and duties of the caregiver and the person being cared for, establishing support measures that promote the maintenance of people being cared for at home [7].

Our review focuses on the training of informal carers and not on the training of community health workers. Although both community health workers and informal carers play crucial roles in supporting the health and well-being of individuals, their nature, responsibility and contexts of action are fundamentally different. The community health worker is a health professional. They are part of the primary health care team and act as a link between the community and health services, while the informal carer is usually a family member, friend or neighbor who provides care for a person who has lost some or all of their autonomy to carry out daily activities [8,9].

The articulation of this Statute with the role of rehabilitation nurses is crucial, as these professionals play a central role in empowering and supporting informal carers, particularly in facilitating care transitions and supporting informal carers, ensuring quality care and preventing overload.

Rehabilitation nurses advocate patient and family involvement in care decisions, aligning with the principles of support and rights for informal carers during care transitions, which aligns with broader principles of carer support [10,11].

ICs face numerous problems, such as relational issues, social restrictions and the demands of caring [12]. Physical barriers and a lack of adaptability at home are some of the most frequently reported problems, even at an early stage of the process.

It is therefore up to the Rehabilitation Nursing Specialist (RNS), who is closest to the dependent person and their family, to know their environmental and physical conditions, as well as the resources available so that help can be improved and, above all, individualized [13]. It is extremely important for the RNS to know the context in which the caregiver and the dependent person live, to be able to identify housing needs and provide technical assistance or existing support in the community [14].

Several studies report evidence of an increase in quality of life associated with the care provided by RNS in various clinical contexts, and particularly in the support provided to informal caregivers (ICs). Rehabilitation nursing improves the health-related quality of life of patients with chronic conditions by addressing intricate care requirements and reinforcing coping strategies. This involvement of informal carers promotes better management of chronic conditions, leading to improved patient outcomes and increased informal carer satisfaction [15,16,17]. ESRs must identify high-risk ICs in order to improve their quality of life by responding to their needs [18].

To provide adequate care, it is necessary to empower ICs with tools that support and facilitate the process and that are appropriate. By improving health literacy, we can positively influence health behaviors and, consequently, promote an improvement in the health status of the individual and the surrounding society [19].

The quality standards for specialized care in rehabilitation nursing issued by the Portuguese Order of Nurses highlight RNS intervention in the training of the person, family and caregiver, with a view to well-being and improving quality of life, as well as the involvement of the client and significant people in the care process, by carrying out teaching and training taking into account the existing resources at home [20]. The RNS’s specific competencies show that they intervene when they teach the person and/or caregiver specific self-care techniques and technologies, and when they discuss the functional alterations presented with the person/caregiver (at motor, sensory, cognitive, cardiac, respiratory, feeding, elimination and sexuality levels) [21].

The aim of this review is to map the most up-to-date scientific evidence on RNS interventions in the IC empowerment of the older person at home.

## 2. Materials and Methods

### 2.1. Study Design

To ensure best practice methods, we use the PRISMA-ScR checklist [Preferred Reporting Items for Systematic Reviews and Meta-Analyses (PRISMA) extension for scoping reviews] [22,23] and the methodology according to the Joanna Briggs Institute (JBI) [24].

This scoping review was not registered, because unlike systematic reviews and meta-analyses that need to be registered in public registers such as PROSPERO, it is not mandatory to register scoping reviews [25].

### 2.2. Inclusion and Exclusion Criteria

We chose to include in this review all types of quantitative or qualitative primary empirical studies, including cross-sectional, longitudinal, observational or experimental studies. This review also included studies with and without a comparison group, and secondary studies such as systematic literature reviews to map gaps in the literature and have a wider range of potential studies to include in the review, limiting the studies to Portuguese, English and Spanish with full text.

In accordance with the JBI recommendations for scoping reviews, inclusion criteria were defined using the PCC method. Therefore, the following PCC was defined for this scoping review:

Participants: informal caregivers of the elderly;

Concept: Rehabilitation Nurse Specialist Interventions;

Context: home.

The exclusion criteria for this scoping review were all articles unrelated to the subject under study, those with dubious methodology, and those duplicated in the research databases.

### 2.3. Search Strategy

The search was conducted in the PubMed and CINHAL Ultimate databases. Based on the problem defined, a list of keywords was drawn up to enable a sufficiently sensitive and specific search. This list was created and validated using MeSH^®^ Browser terms (Medical Subject Headings) and DeCS^®^: Rehabilitation Nursing, Informal Caregivers, Elderly, Aged, Training, Mentoring.

During November 2024, a boolean search (with the Booleans operators “AND” and “OR”) was carried out through PubMed and CINHAL Ultimate, limiting the studies to Portuguese, English and Spanish, from January 2019 to October 2024 and with full text: (Rehabilitation Nursing) AND (Informal Caregivers OR Caregivers) AND (Elderly OR Aged) AND (Mentoring OR Training)). This time limit was considered appropriate given the aim of this review and the search for the most recent literature.

### 2.4. Study Selection Process

The selection of relevant studies was essentially based on the research question, “What are the Interventions of the Rehabilitation Nurse Specialist in the Empowerment of the Informal Caregiver of the Elderly Person at Home?” and not on the assessment of methodological quality since a scoping review seeks to map all the available literature [24].

All duplicate articles were removed using Mendeley (v2.134.0)^®^. Two reviewers independently assessed the inclusion of the studies by reading the titles, abstracts and keywords, excluding those that did not meet the inclusion criteria (Figure 1, showing the PRISMA flowchart). As there was consensus on the articles selected through a face-to-face meeting, it was not necessary to use a third reviewer because the risk of bias was minimized. Subsequently, the full text evaluation phase was carried out using the same method to minimize bias, and it was also not necessary to use a third reviewer using the same procedure.

### 2.5. Quality Assessment of the Studies

The JBI assessment tools were used to analyze the quality of the articles included in this review, which was carried out by the six reviewers independently, with no disagreements. The choice of these specific tools was due to the different methodologies of the studies included as they allowed us to assess the methodological quality of each study and determine if the possibility of bias in its design, conduct and analysis was addressed. The JBI approach considers the best available evidence, the context in which care is provided, the individual patient and the professional judgment and experience of the health care professional [24].

### 2.6. Data Extraction

In the data extraction phase, a descriptive analysis of each study was carried out using a form designed for this purpose. This form was subjected to a pilot test where it was found to be a suitable tool and enabled information to be extracted in accordance with the research question. Data extraction was carried out independently by the six reviewers responsible for selecting the studies, with there being no disagreement. The extracted data contains specific details on the study objective, study design, assessment instruments, participants, interventions and main conclusions.

## 3. Results

The initial search yielded 792 articles. After removing 205 duplicate records and 295 due to the full text not being available, 292 articles remained. Screening of titles and abstracts led to the exclusion of 230 articles that did not meet the inclusion criteria. Of the 62 full articles assessed, 2 could not be retrieved and 53 were excluded for reasons such as focusing on formal carers, not detailing specific RNS interventions or being carried out in institutional settings rather than at home. Finally, seven articles met all criteria and were included in this scoping review.

A total of seven articles were included: two quasi-experimental studies [26,27], a report of experience [28], an observational, cross-sectional, descriptive and quantitative study [29], a systematic review [30], a cross-sectional study [31], and a mixed study [32].

The countries of origin were varied and included the USA, Portugal, Belgium, China and Brazil. In chronological terms, by year of publication or last review, one article corresponded to 2019; two to 2020; one to 2022; and three to 2023. All articles focused on the training of ICs of the older adult at home.

Therefore, this scoping review included the analysis of articles on the training of informal caregivers of the older adult at home, which resulted in the findings shown in Table 1.

Our results show the following:

Rehabilitation programs: two quasi-experimental studies, one in Portugal and the other with Rehabilitation Nurse Specialists, have shown that training programs significantly improve caregivers’ self-care skills (hygiene, transfers, positioning, dressing) and the patients’ autonomy/balance.

Educational technologies: a Brazilian experience report described the development and preliminary positive feedback of an online course for carers of post-stroke elderly people, highlighting the potential of digital tools in nursing education.

Carers’ difficulties: a Portuguese observational study found that carers of more dependent individuals, older carers, and those with architectural barriers face greater difficulties in self-care and activities of daily living.

Impact of caregiver involvement: a systematic review concluded that caregiver participation, through care pathways, education and mediated exercises, can improve functional performance and, secondarily, the quality of life of the elderly, with greater effectiveness in more frequent interventions.

Impact of caregiver involvement: a systematic review concluded that caregiver participation, through care pathways, education and mediated exercises, can improve functional performance and, secondarily, quality of life in the elderly, with greater effectiveness in more frequent interventions.

Quality of home care: a Chinese cross-sectional study revealed that the severity of disability directly affects the quality of home care, but that social support and caregiver competence act as important mediators, suggesting that strengthening these factors improves care.

Caregiver well-being and cost: A hybrid mixed-method study in the US investigated the effectiveness of the ADS Plus program in improving well-being and reducing depression in caregivers of people with dementia. The study highlights that these carers often suffer from depression and burnout due to a lack of adequate education and skills, and the ADS Plus program, adapted to their needs, seeks to mitigate these risks.

In short, the studies emphasize the importance of training and support programs (face-to-face or online) to empower informal carers, reduce their difficulties, and improve the quality of care and the well-being of both carers and dependent older people.

## 4. Discussion

When we talk about informal carers, we are mainly referring to individuals who provide care without a formal employment relationship or specific remuneration for this role. They are typically family members, friends or neighbors who take on the responsibility of caring for someone with a dependency. However, it is important to distinguish these carers from professionals who work in the field of home support or personal care services, even though their day-to-day duties may seem like those of an informal carer. These professionals operate on a formal level, with training, supervision and remuneration [33,34].

It is crucial not to confuse informal carers with:-Family helpers/geriatric assistants/direct action helpers: these are professionals (albeit with different levels of training) who work in home care services, nursing homes or other institutions, and are paid for their services.-Home care nurses, home care physiotherapists, family doctors: these are health professionals with university or technical training who provide specific health care and are paid for it.

The fundamental difference is the formality, training and remuneration associated with the service provided. The informal carer acts out of emotional ties and personal responsibility, not as a profession.

Informal carers of elderly patients with chronic conditions collaborate with a variety of individuals and entities to ensure the well-being of the cared-for person while seeking support for themselves. The main collaborations include health care professionals, family and social networks, and community support services. This multifaceted collaboration is essential to guarantee quality care for the elderly person with a chronic condition, while at the same time caring for the carer and preventing physical and emotional overload for the informal carer.

To formalize informal carers in Portugal, the legislation stipulates that the level of care they will need to provide is characterized by being a permanent carer for a person in a situation of dependency. Formalization aims to recognize and support those who already provide this care on an unpaid basis, be it a family member or, more recently, a non-family member who lives with the person being cared for. Formalizing informal care aims not only to recognize the essential work that these people do but also to provide them with financial support (support allowance for the main informal caregiver), training, rest periods and psychosocial support, contributing to their well-being and improving the quality of life of the person being cared for [35].

The majority of individuals who take on the role of carer often take on a multitude of responsibilities related to the provision of care, without, at the same time, the necessary access to comprehensive health education and the necessary skills to enable them to effectively navigate the myriad of challenges and unpredictable fluctuations inherent in the caring process [32]. The effectiveness of home care is hampered by the lack of pertinent knowledge on the part of informal carers (ICs) in areas such as health, pathologies and pharmacotherapy. Consequently, the support offered does not meet the genuine demands of the individual in a situation of dependency [31]. The acknowledgement of these difficulties is not new, since previous research has already pointed to the same conclusion: adequate training of informal carers is indispensable for the provision of high-quality care [36,37], which can increase the quality of life of ICs that is often diminished by the patient’s clinical circumstances and the patient’s own quality of life [38].

The authors confirm that it is imperative to analyze the difficulties experienced by ICs, the reasons behind them and the effects of training on the quality of care they provide [29]. The authors emphasize that this is a difficulty for the population in general and, even more so, for health professionals. It should be emphasized that clear communication about care needs between the health team responsible and the elderly person’s carer enables the IC to provide the assistance required [30].

One study [26] reported that receiving practical training by ICs through RNS, specifically in techniques for transferring and positioning the elderly, correlated with notable improvements in the quality of care. This improvement culminated in greater safety and comfort for the dependent person. For the authors, practical training is essential to empower carers and mitigate the risk of injury. It is also important to note that the home environment can be a limiting factor since many homes lack the necessary infrastructure to ensure the safety of the elderly, which compromises the effectiveness of the techniques taught by the RNS. [26,29].

A study [29] corroborates that the difficulties faced by ICs, especially in relation to the degree of dependence of the elderly person and architectural barriers, are significant challenges that interrupt the effectiveness of the training. Therefore, a holistic approach to RNS is needed which, among other things, involves adapting the home space to ensure that training can be reproduced safely and effectively. The implementation of devices such as grab rails, articulated beds and health-monitoring systems or automatic alerts are measures that promote the safety of the elderly and optimize the IC’s performance [29]. Evidence from previous research indicates that the spatial configuration of the home plays a crucial role in the patient’s rehabilitation. Older homes, lacking adaptations for addiction, can represent a significant obstacle in this process [39,40].

The body balance training given to the elderly is an example of the direct impact of RNS intervention on the safety, mobility and autonomy of the elderly, reducing, among other things, the risk of falling. This is one of the main problems faced by the elderly, with physical and psychological consequences. The elderly develop a fear of falling, which can limit their willingness to carry out activities and worsen their degree of dependence. It is essential that the RNS trains the IC beyond physical balance, providing emotional support, helping the elderly overcome fears and promoting confidence in their bodies [27]. The combination of practical training and emotional support can be crucial in reducing fear and promoting a more complete recovery [32]. In fact, as previous research has shown, practical training has been shown to be associated with increased levels of peace of mind and quality of life [41,42].

In many cases, the training by the RNS is carried out during older adults’ hospitalization, as is the acquisition of the skills by the IC; so, there may not be enough learning opportunities [30].

A study [28] demonstrates how the use of educational technologies, such as online courses, can be an effective tool for distance CI training, especially in rehabilitation contexts, such as after a stroke. The authors also argue that although digital education is a practical and affordable solution, it is necessary to reflect on accessibility issues. In Portugal, many ICs may not have the necessary level of technological familiarity or access to suitable devices.

Social support is crucial for the effectiveness of training, and one solution would be to create hybrid support programs that integrate digital learning with face-to-face sessions or tutorials led by RNSs. In addition, courses should be interactive and based on practical scenarios to ensure that ICs acquire the knowledge and apply it effectively in a practical context [31]. In fact, we have already mentioned the importance of training ICs, and this training can take many forms, since the key is to empower ICs as much as possible, as reported in previous studies [43,44,45].

The same study also reinforces that social support, whether emotional or practical, is one of the main factors influencing the carer’s competence and, consequently, the quality of care provided to the elderly person.

The training of caregivers by the RNS can be achieved through a variety of interventions, such as formal education, specific skill training and the implementation of mediated exercise programs. These approaches have been proven to induce substantial improvements in the elderly person’s functionality, including improving their ability to get around, minimizing secondary complications, and establishing independence in the home environment. Improved quality of life and faster and more effective recovery are results that have been attested to by the authors [30]. There is consistency between these results and the existing literature, which emphasizes that the application of therapeutic and nursing interventions, including rehabilitation approaches, can generate a notable improvement in the quality of life of patients with chronic illnesses. Such improvement is achieved by focusing on aspects of low performance, which, in turn, improves physical health, promotes independence and contributes to general well-being [46,47].

Social support can take different forms. In addition to emotional support groups and RNS counselling practices, family support and a network of friends play a key role in reducing the burden on the carer [31]. By instituting programs such as the ADS Plus Program, they demonstrate the importance of providing continuous education to the caregiver and families involved [32].

Several studies point to the need for an educational pedagogy to be carried out by the RNS, where its intervention stands out for maximizing the functional capacity of both the IC and the elderly person. Awareness of the IC’s difficulties in their socio-cultural context, and specifically at home, is essential for the creation and application of training programs suited to their real health needs [26,29].

The challenge then arises to rehabilitate, reintegrate and develop active aging policies [29]. Expanding existing community resources with evidence-based programs is a promising approach that can expand the reach and implementation potential of previously scientifically tested interventions [32]. Lectures on health, knowledge of and basic care skills, and an increased number of home visits to the older adult are measures that have been suggested [31]. They also state impact that the attention that political decision-makers should pay to the quality of home care has in the active promotion of a general well-being policy.

The implementation and strengthening of community support programs in Portugal, with the active involvement of RNS—mirroring models already present in some areas—appears to be an effective means of optimizing the well-being and capabilities of ICs, resulting in an improvement in the quality of home care provided to the elderly. It is imperative to point out that intervention programs aimed at CIs of dependent patients have proven to be effective in improving the health status and quality of life of this group, thus justifying the incentive to adopt them on a large scale in primary health care [48].

Analyzing the literature reveals that the quality of home care is intrinsically linked to the training and support of ICs, a multifaceted challenge that persists in public health. The lack of technical–scientific knowledge about health, pathologies and pharmacotherapy limits ICs’ ability to meet the real needs of dependent elderly people, creating a mismatch between the care provided and the support required. This problem is exacerbated by the inadequacy of many home environments and the lack of effective communication between health professionals and ICs.

However, studies show that targeted interventions, especially practical training and emotional and social support, are crucial to mitigating these difficulties. The involvement of the RNS in education and the teaching of skills, such as transfer and positioning techniques, has a direct impact on the safety, mobility and autonomy of the elderly, reducing risks such as falls and promoting greater well-being and independence. The implementation of educational technologies and the creation of hybrid support programs, which combine digital learning with face-to-face sessions, show promise for overcoming access barriers and ensuring the applicability of skills. In addition, social support, both formal (support groups, RNS counselling) and informal (family and friends), is a determining factor in reducing carer burden and improving the quality of care.

Given this evidence, it is recommended that community support programs be implemented and strengthened in Portugal, with the active participation of RNS. These programs should focus on the comprehensive training of ICs, incorporating practical training, continuing education, emotional support and the development of communication skills. It is essential that health policies consider the reality of the home, encouraging architectural adaptations and the use of safety technologies. The expansion of intervention programs for ICs in primary health care, which have been proven to improve the health status and quality of life of both the caregiver and the elderly person, is an imperative step towards promoting a general well-being policy and ensuring high-quality home care for the dependent elderly population.

In summary, the findings reinforce the need for integrated strategies that combine technical training, emotional support and home adaptations to promote quality care. The proactive role of the Nursing Service, combined with public policies focused on training and supporting carers, is essential to guarantee dignity and well-being for the elderly who receive care at home.

Table 2 summarizes the different types of interventions that can be adopted by the RNS.

## 5. Conclusions

The RNS’s interventions in IC training play a crucial role in improving the quality of life of dependent older adults at home. Empowering caregivers to carry out essential practical tasks, such as performing daily activities or body balance, promotes the autonomy of the older adult and guarantees their safety and well-being.

The use of innovative approaches, such as online courses and community support programs, has been shown to be effective in improving caregivers’ skills while offering ongoing emotional and educational support. This contributes to reducing the emotional burden on caregivers and improving the quality of care provided, which is directly reflected in an increase in the older adult’s quality of life.

The identification of predictive variables, such as the functional dependence of the elderly person or architectural barriers, implies a holistic and personalized approach by the RNS.

The RNS’s interventions in IC training are fundamental to guaranteeing the safety, autonomy and well-being of dependent older adults at home. These interventions, set out in the legal and regulatory profession framework, improve the caregiver’s skills, reduce emotional overload, and increase the effectiveness of care. It is therefore essential that caregiver training programs, created and developed by RNS, are continuous, personalized and evidence-based, ensuring that caregivers are properly prepared to meet the needs of the older adult at home.

Future research is recommended to assess the long-term impact of RNS interventions on the quality of life of older people and the burden on carers, identify additional predictive variables for the success of interventions, compare the effectiveness of different training approaches (online, community-based), and analyze the cost–benefit of training programs. It is also recommended that health policymakers ensure continued funding for RNS training programs, integrate caregiver training into aging policies, and expand and ensure accessibility to programs in all regions.

## 6. Study Limitations

This scoping review only included studies in English, Portuguese and Spanish; so, there may be a linguistic bias. Likewise, only studies from the last 5 years were included, and studies without an abstract or full article were also excluded, which may have limited the study.

## Figures and Tables

**Figure 1 ijerph-22-00971-f001:**
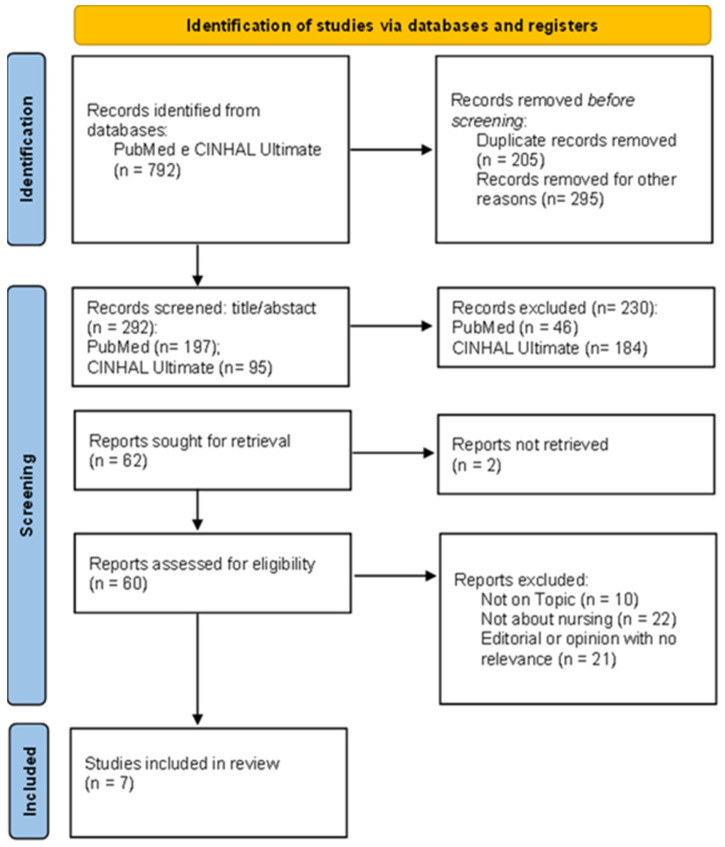
PRISMA flow diagram of the study selection process.

**Table 1 ijerph-22-00971-t001:** Description of the included studies.

Reference	Objectives	Methodology	Results
[26]	To evaluate the effects of a rehabilitation nursing program on the training of informal caregivers in the self-care of the elderly dependent due to stroke at home.	Quasi-experimental study. The sample consisted of 15 informal caregivers from a Community Care Unit in northern Portugal. The variables assessed before and after the implementation of the program were as follows: caring for personal hygiene, transferring, positioning, providing technical aids, using the toilet, feeding/hydrating and dressing/undressing. The program included six contacts based on teaching, instruction and skills training. Data collection instruments used: the Informal Caregiver Characterization Form and the Informal Caregiver Capacities Scale for Elderly Dependent on Stroke. Keywords: caregivers, stroke, elderly, self-care, rehabilitation nursing.	Most of the caregivers in the sample were women and on average 59.9 years old. In all areas of self-care, there was an improvement in their ability after the intervention, and it was more significant in those which initially presented greater difficulty: dressing/undressing, transferring, positioning. The rehabilitation nursing program favorably influenced the training of informal caregivers for self-care of the elderly dependent on stroke at home. This research provides support to health teams for meaningful clinical practice for populations, corroborating the fundamental role of individualized intervention by rehabilitation nurses.
[27]	To evaluate the contribution of a training program, applied by Specialists in in Rehabilitation Nursing, to informal caregivers to address body balance in the dependent person in a home context.	Quasi-experimental, single-group, longitudinal study with observation before and after the training program, in a home visit context, with the application of a program divided into six sessions. This was a non-probabilistic, convenience sample with 10 informal caregivers and their dependent loved ones. A sociodemographic form was used to collect the data, a search in the patient’s computerized clinical file was conducted, and an observation grid was drawn up for the purpose, using the Tinetti Test scale and the Barthel Index. Keywords: rehabilitation nursing, postural balance, informal caregiver, training, elderly.	The study highlights that training informal caregivers can result in significant improvements in the balance and autonomy of dependent people at home. It showed that, after the EER intervention, informal caregivers acquired theoretical and practical knowledge about body balance, and the patients showed significant average gains in balance and autonomy. This suggests that people with a high degree of dependency can benefit from basic care related to static sitting balance, provided their caregivers are trained to do so. The results of the study reinforce the importance of thinking about and implementing more advanced Rehabilitation Nursing programs and/or programs with higher objectives, to be developed in the context of home visits, thus contributing to the training of the informal caregiver and health gains for the dependent person.
[28]	To describe the process of developing a massive, open, online course for informal caregivers of elderly people with a medical diagnosis of stroke.	Experience report for the construction and development of a massive, open and online course for informal caregivers of elderly people who have suffered a stroke admitted to a hospital in southern Brazil, written by nurses and a digital programmer. Keywords: educational technology, nursing education, stroke.	The process of developing a massive, open and online course requires a team with expertise in different areas (nursing and digital). The course had a positive preliminary evaluation from the target audience (three informal caregivers) regarding its content and functionality and represents an important advance for nursing in the construction of digital educational technologies. They recommend the development of further courses to assess their accessibility and functionality with a larger audience.
[29]	To assess the difficulties faced by informal caregivers in caring for dependent people and to identify variables that predict these difficulties.	This is an observational, cross-sectional, descriptive–correlational and quantitative study that used a non-probabilistic convenience sample of 119 ICs from the Central Region of Portugal. The measuring instrument used included a sociodemographic data sheet and an Informal Caregiver Difficulties Assessment Scale (EADCI). Key words: caregiver, learning disabilities, home patients, rehabilitation nursing.	The authors observed difficulties, especially in the dimensions of self-care and activities of daily living. Difficulties are more pronounced among caregivers of people with severe dependence, older caregivers and those who face architectural barriers at home. The predictive variables thus identified were the degree of functional dependence of the dependent person, the age of the caregiver, and the existence of architectural barriers in the home. Informal caregivers present difficulties at various levels of care, highlighting the need to implement new strategies to respond to these challenges.
[30]	To determine whether the involvement of the caregiver in hospital or after discharge can increase the functional performance of the elderly. The secondary objective was to determine whether the involvement of the caregiver can influence the quality of life of the patient and the caregiver.	Systematic review with narrative synthesis. MEDLINE, Embase, CINAHL, Cochrane and Web of Science databases were searched for (quasi-)experimental and observational studies, with the following inclusion criteria: caregiver involvement in functional functional performance, average age over 65, admitted to a hospital unit and subsequently discharged to their home environment. Key words: caregiver involvement, older adults, functional performance, hospitalization, physiotherapy.	Three types of interventions for caregivers could be distinguished: a care pathway (inclusion of caregivers in the care process), stroke education and teaching of bedside management skills, and exercises mediated by the caregiver. The only study that evaluated the care pathway recorded 24.9% more returns home in the intervention group. The studies that evaluated the effect of education and teaching bedside handling skills reported higher levels of efficacy for various outcomes as the frequency of sessions increased. All studies with caregiver-mediated exercises showed beneficial effects on functional performance immediately after the intervention and at 3-month follow-up.
[31]	To explore the relationship between the variables “severity of disability”, “social support” and “caregiver competence”, and the quality of home care in a population of elderly Kazakh people in a rural area of China.	This was a cross-sectional study of 335 disabled elderly people and their main informal caregivers. The severity of the disability was assessed using the Activities of Daily Living Scale (Katz Index). Caregiver competence was assessed using the Family Caregiver Task Inventory. Social support was assessed using the Social Support Rating Scale. The quality of home care was assessed using the Family Caregiver Consequences Inventory Scale. Keywords: structural equation model, severity of disability, quality of home care, elderly Kazahk population with disabilities, primary family caregiver.	It was shown that the severity of the disability had a direct effect of 29.28% on the quality of home care and an indirect effect of 70.72% through social support and caregiver competence. The results confirm that the better the social support and the competence of the caregiver, the better the quality of home care available to elderly people with disabilities. Policymakers should prioritize improving the quality of care provided to older people with severe disabilities. Health care management departments should provide training for informal caregivers to improve their knowledge and skills in health care and rehabilitation.
[32]	This study has two main objectives: - To evaluate the effectiveness of ADS Plus in improving caregiver well-being and reducing depressive symptoms, compared to the existing ADS program, at 6 months. - To evaluate the effects of maintaining ADS Plus for 12 months on caregiver well-being and the impact on depressive symptoms. Secondary objectives: - To assess whether caregivers who receive the ADS Plus program are more likely to keep family members in the ADS program and less likely to place their family members with dementia in residential settings compared to those in routine ADS over 12 months. - To estimate the costs of ADS Plus and assess whether it results in net financial benefits when compared to usual ADS at 6 and 12 months. - To assess the effects of ADS Plus on the behavioral symptoms of people living with dementia and on caregiver efficacy and concern about managing these symptoms.	Dementia; neuropsychological behaviors; family caregiving; activities; occupational therapy; psychosocial intervention. A total of 30 ADS sites in the United States and 300 family caregivers. Hybrid design involving a two-group clinical trial to evaluate the program’s effectiveness. Mixed methodologies to evaluate the program’s implementation processes.	Caring for an individual living with dementia can put caregivers at risk of depression, burnout, health problems, and financial burdens. Most caregivers take on responsibilities without access to education about the disease or the skills needed to manage the challenges of caregiving. Expanding existing community-based resources, such as adult day centers, with evidence-based programs designed to support family members in their caregiving efforts is a promising approach. The ADS Plus program is tailored to the caregiver’s needs. In the program, the caregiver initially identifies three to five problem areas they want to address. Over 12 months, caregivers learn a specific approach to managing challenges. ADS Plus also provides ongoing education about dementia, referrals, and connections (if needed) to address unmet needs, as well as a prescription for each problem area. These “prescriptions” offer specific strategies for dealing with a challenge identified by the caregiver. The strategies provided include communication, task simplification, environmental adjustments, and self-care. If ADS Plus proves effective, it suggests that expanding services for older adults could be a potential model for scaling up support programs for caregivers. The importance of this trial is underscored by the high rates of depression among dementia caregivers, the projected increase in new dementia cases in the coming decades, and the critical role that community programs, such as adult day services, play in supporting families living with dementia.

**Table 2 ijerph-22-00971-t002:** RNS interventions.

RNS Interventions	
Teaching about transfers and positioning	Balance and mobility training
Teaching about the administration of medication, control of chronic diseases and prevention of complications	Promoting appropriate physical activities
Home’s assessment and identification of architectural barriers that may disturb care and mobility	Implement training courses to empower ICs
Recommendations on home adaptations, such as the installation of support bars and articulated beds	Promoting access to and use of appropriate technologies and devices
Emotional support in relation to falls and encouraging autonomy	Encouraging participation in existing support groups and family networks
Regular evaluation of the care provided by the IC	Adjust IC training according to needs and difficulties

## Data Availability

The original contributions presented in this study are included in the article. Further inquiries can be directed to the corresponding author.

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
