# Peer review of "Interventions by Rehabilitation Nurse Specialists in the Training of Informal Carers of Older People at Home with Chronic Diseases: A Scoping Review"

_ijerph, 2025, doi:10.3390/ijerph22070971_

Round 1
Reviewer 1 Report
Comments and Suggestions for Authors
Introduction
The introduction of the article presents a solid and up-to-date thematic foundation, based on demographic and social data to justify the relevance of the study. The concern with population aging is contextualized both globally and nationally, using data from the 2021 Portuguese census to demonstrate the magnitude of the issue. This approach is pertinent and helps situate the reader regarding the urgency of the topic.
However, there are formal and structural aspects that could be improved. Firstly, there is some redundancy and repetition that hinders the text’s flow, such as in: “improving patient coping strategies and patient coping strategies” and “ultimately leading to better ultimately leading to better management.” These errors suggest incomplete revision or a lack of attention in the final editing. In addition, the repeated use of “with chronic” in line 71 should be corrected. Although minor, these slips compromise the clarity and professionalism of the text.
The central argument — that rehabilitation specialist nurses (RNS) play an essential role in supporting informal caregivers (IC) — is coherent and well supported by scientific literature. However, some passages could be better organized to avoid dispersion. For example, when introducing the Portuguese legislation on the Informal Caregiver Statute, it would be appropriate to briefly elaborate on how this public policy connects to the role of the RNS, establishing a clearer bridge to the study’s objective.
Methodology
Inclusion and Exclusion Criteria (lines 100–112)
There is inconsistency in stating that "all types of primary and secondary studies" were included, without justifying the relevance of systematic reviews or secondary studies in a scoping review.
Suggestion: Specify which types of secondary studies were included and for what purpose (e.g., to map literature gaps).
Study Selection (lines 124–135)
The absence of a third reviewer should be better justified.
Suggestion: Add a sentence explaining how consensus was reached or why the risk of bias was considered low.
Data Extraction and Analysis (lines 138–143)
The extraction tool is not sufficiently described.
Suggestion: Specify whether a standardized form was used, whether it was pilot-tested, and include it as supplementary material if possible.
Results
Clarity and Scientific Writing
Issue: Phrases such as “After removing 205 duplicated and 295 for other reasons” lack precision and fluency.
Suggestion: Write: “After removing 205 duplicate records and 295 due to ineligibility based on title or abstract screening...”
Issue: Redundancy in passages such as “60 articles remained. 53 were excluded... 7 articles met all criteria.”
Suggestion: Consolidate: “Of the 62 full-text articles assessed, 2 could not be retrieved and 53 were excluded for reasons such as...”
Discussion
Coherence and Textual Flow
Issue: There are long and dense sections with conceptual repetitions (e.g., reiterating the importance of IC training).
Solution: Group similar ideas, synthesize repetitive phrases, and avoid redundancies such as “as previously mentioned” when the idea is still under development.
Example Rewrite:
"On several occasions, the literature points out that practical training for informal caregivers is essential to ensure safety and comfort for the older adult. In addition to physical support, emotional training significantly contributes to strengthening the self-confidence of both the caregiver and the patient."
Paragraph Connection
Issue: Some paragraphs seem disconnected or abruptly change focus (e.g., jumping from practical training to architectural barriers).
Solution: Insert transition sentences and establish logical connections between topics.
Example Transitional Sentence:
"In addition to the caregiver’s technical training, the home environment must be considered a facilitator or barrier to rehabilitation, especially regarding architectural obstacles."
Precision and Language Clarity
Issue: Occasional use of vague terms or unclear constructions (“reproduced safely,” “many homes are not prepared...”).
Solution: Replace with more objective and clear expressions.
Example:
"The absence of structural adaptations, such as support bars or articulated beds, compromises the safe application of the techniques taught during training."
Conclusion of the Discussion
Issue: The section ends abruptly, without a final synthesis or clear practical/political implications.
Solution: Conclude with a paragraph that reinforces the main findings and recommends future actions.
Example Conclusion:
"In summary, the findings reinforce the need for integrated strategies combining technical training, emotional support, and home adaptations to promote quality care. The proactive role of RNS, together with public policies focused on caregiver training and support, is essential to ensure dignity and well-being for older adults receiving care at home."
Author Response
Dear Reviewer 1,
Thank you very much for your very helpful comments. We will respond to them and we think that the manuscript has been improved by your suggestions. We will respond in order to improve the manuscript. Please let us know if any further changes are needed.
Comment 1:
Introduction
The introduction of the article presents a solid and up-to-date thematic foundation, based on demographic and social data to justify the relevance of the study. The concern with population aging is contextualized both globally and nationally, using data from the 2021 Portuguese census to demonstrate the magnitude of the issue. This approach is pertinent and helps situate the reader regarding the urgency of the topic.
However, there are formal and structural aspects that could be improved. Firstly, there is some redundancy and repetition that hinders the text’s flow, such as in: “improving patient coping strategies and patient coping strategies” and “ultimately leading to better ultimately leading to better management.” These errors suggest incomplete revision or a lack of attention in the final editing. In addition, the repeated use of “with chronic” in line 71 should be corrected. Although minor, these slips compromise the clarity and professionalism of the text.
The central argument — that rehabilitation specialist nurses (RNS) play an essential role in supporting informal caregivers (IC) — is coherent and well supported by scientific literature. However, some passages could be better organized to avoid dispersion. For example, when introducing the Portuguese legislation on the Informal Caregiver Statute, it would be appropriate to briefly elaborate on how this public policy connects to the role of the RNS, establishing a clearer bridge to the study’s objective.
Response 1:
Thank you very much for your comment. We've reworded parts of the introduction. Lines 84-90; 102-106.
Comment 2:
Methodology
Inclusion and Exclusion Criteria (lines 100–112)
There is inconsistency in stating that "all types of primary and secondary studies" were included, without justifying the relevance of systematic reviews or secondary studies in a scoping review.
Suggestion: Specify which types of secondary studies were included and for what purpose (e.g., to map literature gaps).
Response 2:
Thank you very much for your pertinent comment. We've added lines 135-137.
Comment 3: Study Selection (lines 124–135)
The absence of a third reviewer should be better justified.
Suggestion: Add a sentence explaining how consensus was reached or why the risk of bias was considered low.
Response 3:
Thank you very much for your comment. It is indeed necessary to clarify. We've reworded lines 166-170.
Comment 4: Data Extraction and Analysis (lines 138–143)
The extraction tool is not sufficiently described.
Suggestion: Specify whether a standardized form was used, whether it was pilot-tested, and include it as supplementary material if possible.
Response 4:
Thank you very much for your comment. We've reworded lines 184-187 to clarify.
Comment 5:
Results
Clarity and Scientific Writing
Issue: Phrases such as “After removing 205 duplicated and 295 for other reasons” lack precision and fluency.
Suggestion: Write: “After removing 205 duplicate records and 295 due to ineligibility based on title or abstract screening...”
Response 5:
Thank you very much for your comment. We agree with your suggestion and have clarified lines 192-193.
Comment 6: Issue: Redundancy in passages such as “60 articles remained. 53 were excluded... 7 articles met all criteria.”
Suggestion: Consolidate: “Of the 62 full-text articles assessed, 2 could not be retrieved and 53 were excluded for reasons such as...”
Response 6:
Thank you very much for your pertinent comment. We agree with your suggestion and have clarified lines 194-197.
Comment 7:
Discussion
Coherence and Textual Flow
Issue: There are long and dense sections with conceptual repetitions (e.g., reiterating the importance of IC training).
Solution: Group similar ideas, synthesize repetitive phrases, and avoid redundancies such as “as previously mentioned” when the idea is still under development.
Example Rewrite:
"On several occasions, the literature points out that practical training for informal caregivers is essential to ensure safety and comfort for the older adult. In addition to physical support, emotional training significantly contributes to strengthening the self-confidence of both the caregiver and the patient."
Paragraph Connection
Issue: Some paragraphs seem disconnected or abruptly change focus (e.g., jumping from practical training to architectural barriers).
Solution: Insert transition sentences and establish logical connections between topics.
Example Transitional Sentence:
"In addition to the caregiver’s technical training, the home environment must be considered a facilitator or barrier to rehabilitation, especially regarding architectural obstacles."
Precision and Language Clarity
Issue: Occasional use of vague terms or unclear constructions (“reproduced safely,” “many homes are not prepared...”).
Solution: Replace with more objective and clear expressions.
Example:
"The absence of structural adaptations, such as support bars or articulated beds, compromises the safe application of the techniques taught during training."
Conclusion of the Discussion
Issue: The section ends abruptly, without a final synthesis or clear practical/political implications.
Solution: Conclude with a paragraph that reinforces the main findings and recommends future actions.
Example Conclusion:
"In summary, the findings reinforce the need for integrated strategies combining technical training, emotional support, and home adaptations to promote quality care. The proactive role of RNS, together with public policies focused on caregiver training and support, is essential to ensure dignity and well-being for older adults receiving care at home."
Response 7:
Thank you very much for your comments. On the basis of your comments we have revised the whole discussion in an attempt to improve it. We think it's more appropriate now. If the reviewer feels that any further changes are necessary, please let us know.
Best regards.
Reviewer 2 Report
Comments and Suggestions for Authors
Thank you for the opportunity to review this work. The authors undertook an important review, however, they are not situating their work in a growing body of literature. Below, I provide some comments which may help them to refine the review further.
Title: I am struggling to identify the context of the review. Are the authors mapping interventions in low-, middle- and high-income countries?
The abstract is well written.
Introduction
The authors provided a good description of Portugal's healthcare system, policy innovations, needs of elderly patients, etc. Only one sentence refers to the outside world. We have a growing body of literature focused on community health workers, many countries are using this cadre of lay health workers to provide basic health care to the elderly with and without chronic conditions. The authors are encouraged to situate their work within this exciting body of literature. The review must be relatable to readers outside Portugal.
Methods
Lines 101-103: Why did the authors included quantitative, qualitative, including primary empirical studies, cross-sectional and longitudinal studies? A justification is required.
The inclusion criteria are plain. Add more details
- Participants – what about education/training level, and gender?
- Context of the review – we have low-, middle- and high-income countries. Does your review cover all these countries? This is not coming out clearly.
- Year – specify the timeframe of the review and explain why.
- As I mentioned earlier, many countries use community health workers to care for the elderly. I am concerned might have included studies reporting on community health workers, as in some contexts they are referred to as informal health workers, etc. I strongly recommend you reflect on the work of community health workers in the introduction and attempt to differentiate them from informal caregivers. Under the exclusion criteria explain studies reporting on community health workers will be excluded.
Search strategy
Why did you limit the search to PubMed and CINHAL. Please provide reasons.
The search years 2019 to 2024 need justification.
Lines 161-163 specify the range of interventions provided by RNS in empowering ICs. Consider unpacking these interventions in detail. These interventions include caregiver education and training, psychosocial support, promotion of caregiver self-care, and use of technology. Kindly unpack these interventions before discussing them.
Discussion
The discussion suffers from the gaps identified in the introduction. The authors are encouraged to read further about informal health care systems. Countries have different models of informal healthcare systems. I gave an example of community health workers; in many countries, community health workers are already trusted in providing health support to the elderly. For example, in South Africa, they visit the elderly every week to provide basic care, support medication adherence, etc. Before they were recruited to join the healthcare sy stem, they were called informal carers or helpers. The authors need to think of:
- What other workers are operating at the level of the informal caregivers?
- Who do the informal caregivers collaborate with in caring for elderly patients with chronic conditions?
- If we are to formalise the informal carers, at what level will they need to provide care?

Author Response
Dear Reviewer 2,
Thank you very much for your very useful comments. We will respond to them and we think that the manuscript has been improved by your suggestions. We will respond in order to improve the manuscript. Please let us know if any further changes are needed.
Comment 1:
Title: I am struggling to identify the context of the review. Are the authors mapping interventions in low-, middle- and high-income countries?
The abstract is well written.
Response 1:
Thank you for your comment. We didn't choose to specify between low-, middle- and high-income countries, but we understand the meaning of your comment.
Comment 2:
Introduction
The authors provided a good description of Portugal's healthcare system, policy innovations, needs of elderly patients, etc. Only one sentence refers to the outside world. We have a growing body of literature focused on community health workers, many countries are using this cadre of lay health workers to provide basic health care to the elderly with and without chronic conditions. The authors are encouraged to situate their work within this exciting body of literature. The review must be relatable to readers outside Portugal.
Response 2:
Thank you very much for your comment. We've added lines 43-56 for the sake of clarity.
Comment 3:
Methods
Lines 101-103: Why did the authors included quantitative, qualitative, including primary empirical studies, cross-sectional and longitudinal studies? A justification is required.
Response 3:
Thank you very much for your comment. We have justified it by adding lines 135-136.
Comment 4: The inclusion criteria are plain. Add more details
Response 4:
Thank you very much for your comment. We've added lines 136-137.
Comment 5: Participants – what about education/training level, and gender?
Response 5:
Thank you very much for your comment. We chose not to specify the level of education/training and gender, but we understand your comment
Comment 6:
Context of the review – we have low-, middle- and high-income countries. Does your review cover all these countries? This is not coming out clearly.
Response 6:
Thank you very much for your comment. We chose not to specify between low-, middle- and high-income countries, but we understand the meaning of your comment.
Comment 7: Year – specify the timeframe of the review and explain why.
Response 7:
Thank you very much for your comment. We have clarified this issue by adding lines 156-157.
Comment 8: As I mentioned earlier, many countries use community health workers to care for the elderly. I am concerned might have included studies reporting on community health workers, as in some contexts they are referred to as informal health workers, etc. I strongly recommend you reflect on the work of community health workers in the introduction and attempt to differentiate them from informal caregivers. Under the exclusion criteria explain studies reporting on community health workers will be excluded.
Response 8:
Thank you very much for your pertinent comment. It is indeed necessary to clarify. We've added lines 76-83.
Comment 9:
Search strategy
Why did you limit the search to PubMed and CINHAL. Please provide reasons.
Response 9:
Thank you very much for your comment. We limited the search to these databases because of the quality and relevance of the content and the scope within the health area, but we understand your comment.
Comment 10: The search years 2019 to 2024 need justification.
Response 10:
Thank you very much for your comment. We've added lines 156-157.
Comment 11: Lines 161-163 specify the range of interventions provided by RNS in empowering ICs. Consider unpacking these interventions in detail. These interventions include caregiver education and training, psychosocial support, promotion of caregiver self-care, and use of technology. Kindly unpack these interventions before discussing them.
Response 11:
Thank you very much for your comment. We've chosen to delete lines 199-202 to make it less confusing for the reader, if the proofreader agrees.
Comment 12:
Discussion
The discussion suffers from the gaps identified in the introduction. The authors are encouraged to read further about informal health care systems. Countries have different models of informal healthcare systems. I gave an example of community health workers; in many countries, community health workers are already trusted in providing health support to the elderly. For example, in South Africa, they visit the elderly every week to provide basic care, support medication adherence, etc. Before they were recruited to join the healthcare sy stem, they were called informal carers or helpers. The authors need to think of:
What other workers are operating at the level of the informal caregivers?
Who do the informal caregivers collaborate with in caring for elderly patients with chronic conditions?
If we are to formalise the informal carers, at what level will they need to provide care?
Response 12:
Thank you very much for your pertinent comments. On the basis of your comments we have revised the whole discussion in an attempt to improve it. We think it is now more appropriate. If the reviewer feels that any further changes are necessary, please let us know.
Best regards.
Reviewer 3 Report
Comments and Suggestions for Authors
This scoping review maps the existing literature on interventions by rehabilitation nurse specialists aimed at empowering informal caregivers of older adults at home. Thank you for this review. It is really important to increase family caregivers comfort and confidence to provide care effectively at home because worldwide family caregivers are the largest care workforce. The review follows rigorous methodological frameworks, including the PRISMA-ScR guidelines and the Joanna Briggs Institute (JBI) recommendations, ensuring transparency and systematicity in its process.
Methods: The authors conducted search in 2 databases: PubMed and CINAHL databases, utilizing MeSH and DeCS terms related to rehabilitation nursing, informal caregiving, and the elderly, limited to studies published from 2019 to 2024 in Portuguese, English, and Spanish. The inclusion criteria embraced all empirical primary studies—quantitative, qualitative, or mixed methods—focused on interventions in home settings, targeting informal caregivers of older adults, with no restriction on comparison groups. This inclusive approach enhances the broadness of the evidence map but also introduces heterogeneity among study designs and contexts.
The selection process was conducted independently by two reviewers with consensus procedures, minimizing selection bias. The data extraction process was similarly independent, capturing detailed information about study objectives, methodologies, interventions, and outcomes. The quality assessment employed JBI tools, which are suited for evaluating methodological robustness across diverse study designs, underscoring the authors' commitment to rigor despite the scoping review's map-oriented nature.
Findings: The findings included seven articles from various countries—USA, Portugal, Belgium, China, and Brazil—spread across publication years 2019 to 2023. These studies primarily examined interventions such as caregiver education and training, psychosocial support, promotion of self-care, and telehealth technologies.
The review underscores recognizing the important role rehabilitation nurses can play in empowering informal caregivers within home environments. It highlights that these interventions need to be person-centered, multi-component and address educational, psychosocial, and technological aspects.
Overall, this scoping review can offer a valuable synthesis of recent efforts to support informal caregivers through targeted nursing interventions, emphasizing the importance of this role in aging healthcare systems. It is an important study to publish. However, the readability and reporting can be improved before it can be published in this high impact journal.
Improving the article
Readability can be improved by good ordering. You want to tell your reader what to expect and put it an order in which they would expect it. So in Study Design: you want to tell your reader that you are doing a scoping review, why you chose a scoping review rather than a rapid review, narrative reviews, systematic reviews, integrative reviews, or other type of literature review. Then describe your study design in order. Did you really do 2.5 Data Extraction before 2.6 quality assessment of the study? Similarly, in the results section, you talk about analysis of the articles (line 161) and then that you include 7 articles (Line 165).
Similarly in the Discussion section, you want to discuss your all of your findings in an order that your reader expects. You speak to 4 themes: 1) caregiver education and training, 2) psychosocial support, 3) promotion of self-care, and 4) telehealth technologies. You speak to 1 then 4, then 2, move back to 1 then to 2.
Results Reporting
You have an excellent table about the results, but you do need to summarize your results in the results section. Your reader should not have to extract the what the 7 articles said about your 4 themes: 1) caregiver education and training, 2) psychosocial support, 3) promotion of self-care, and 4) telehealth technologies.
Discussion
I like how you have discussed your findings in relation to the literature. If you had a summary of the literature you reviewed, it would have been easier for your reader to understand what you included in your discussion (why you discuss what you did).
One of my professors said, you want to take your reader by the hand from the first sentence. The introduction should lead to the need for the study, the methods how you did the study, the results tell you what you find, and then you discuss your results in relation to the other relevant literature.
Language
I think it would be good to remove the ageist and overly judgemental stereotyping language. Stereotyping refers to making generalizations about a group of people and applying them to every individual within that group. This can lead to negative judgments and discrimination. Prejudice is a preconceived opinion, usually negative, that is not based on reason or actual experience.
e.g., ageist page 2 line 46 to 49
Original:
When we talk about an ageing population, we think of people who are sicker, with associated comorbidities, dependent and with a gradual loss of functional capacity, and therefore with greater health care needs.
Could be Revised:
When discussing an aging population, it’s important to recognize the diversity of health experiences. While some individuals may experience health challenges, including comorbidities or changes in functional abilities, aging does not inherently equate to dependency or uniform decline. Healthcare needs can vary widely and are influenced by individual circumstances, accessibility to preventative care, and societal support systems.
e.g., overly judgemental, stereotypical language Pg 8 -9 Lines 178 to 196
Original
"The quality of home care is not achieved, since the care provided by IC does not meet the real needs of the dependent person due to the lack of relevant knowledge about health, illness, drug therapy, among others[25]. These difficulties are in line with what has been reported in previous studies, as the training of ICs is crucial if quality care is to be provided [27,28], which can increase the quality of life of ICs that is often diminished by the patient's clinical circumstances, and the quality of life of the patient themselves [29]. The authors confirm the need to study the difficulties experienced by IC, their causes and the impact of IC training on the care they provide [23]. They state that this is a challenge for the population in general and health professionals in particular. Good communication about care needs between the RNS and the caregiver of the older person makes the IC capable to provide the necessary support [24]. One study [20] emphasised that ICs who received practical training through the RNS on techniques such as transferring and positioning the older adult showed significant improvements in the quality of care, resulting in greater safety and comfort for the person.
For the authors, practical training is essential to empower caregivers and reduce the risk of injury. It is also important to emphasise that the domestic reality can be a challenge, as many homes are not prepared to offer the necessary safety to the older adult, which limits the effectiveness of the techniques taught by the RNS [20,23] .
Could be revised:
"Providing quality home care for dependent individuals can be challenging for informal caregivers (ICs), who often take on this role with immense dedication but may lack access to specialized training in areas like health management, illness care, or safe drug administration [25]. Without adequate support, caregivers may face difficulties aligning their care with the evolving needs of their loved ones. Research emphasizes that empowering ICs with tailored education and training—such as guidance on safe patient transfer techniques, home safety adaptations, or stress management—can enhance both the caregiver’s confidence and the care recipient’s quality of life [27–29].
Collaboration between healthcare professionals (e.g., RNS teams) and caregivers is critical. For example, studies show that practical, hands-on training from RNS professionals significantly improves caregivers’ ability to provide safe, comfortable care while reducing risks like injury [20,23]. However, challenges persist, such as home environments that may not easily accommodate safety modifications, highlighting the need for creative, individualized solutions [20,23].
Ultimately, supporting ICs through accessible training, open communication with healthcare teams, and systemic recognition of their invaluable role can help them thrive in their caregiving journey while ensuring the best outcomes for those they care for [24]."
Key changes to reduce stigma/offense:
- Acknowledges caregivers’ efforts: Phrases like "immense dedication" and "invaluable role" validate their commitment.
- Focuses on systemic gaps: Attributes challenges to lack of access to training (not caregivers’ shortcomings).
- Uses empowering language: "Empowering," "collaboration," and "supporting" frame caregivers as partners, not obstacles.
- Balances challenges with solutions: Highlights research-backed strategies (training, teamwork) rather than solely critiquing care quality.
This approach preserves the study’s findings while centering respect for caregivers’ humanity and efforts.
All in all, this will be an excellent scoping review with revisions.

Author Response
Dear Reviewer 3,
Thank you very much for your very useful comments. We will respond to them and we think that the manuscript has been improved by your suggestions. We will respond in order to improve the manuscript. Please let us know if any further changes are needed.
Comment 1:
Improving the article
Readability can be improved by good ordering. You want to tell your reader what to expect and put it an order in which they would expect it. So in Study Design: you want to tell your reader that you are doing a scoping review, why you chose a scoping review rather than a rapid review, narrative reviews, systematic reviews, integrative reviews, or other type of literature review. Then describe your study design in order. Did you really do 2.5 Data Extraction before 2.6 quality assessment of the study?
Response 1:
Thank you very much for your pertinent comment. We have reordered sections 2.5 and 2.6.
Comment 2: Similarly, in the results section, you talk about analysis of the articles (line 161) and then that you include 7 articles (Line 165).
Response 2:
Thank you very much for your comment. We've chosen to delete lines 199-202 to make it less confusing for the reader, if the proofreader agrees.
Comment 3: Similarly in the Discussion section, you want to discuss your all of your findings in an order that your reader expects. You speak to 4 themes: 1) caregiver education and training, 2) psychosocial support, 3) promotion of self-care, and 4) telehealth technologies. You speak to 1 then 4, then 2, move back to 1 then to 2.
Response 3:
Thank you very much for your pertinent comment. On the basis of your comments, we have revised the whole discussion in an attempt to improve it. We think it is now more appropriate. If the reviewer feels that any further changes are necessary, please let us know.
Comment 4:
Results Reporting
You have an excellent table about the results, but you do need to summarize your results in the results section. Your reader should not have to extract the what the 7 articles said about your 4 themes: 1) caregiver education and training, 2) psychosocial support, 3) promotion of self-care, and 4) telehealth technologies.
Response 4:
Thank you very much for your comment. We've added lines 214-243.
Comment 5:
Discussion
I like how you have discussed your findings in relation to the literature. If you had a summary of the literature you reviewed, it would have been easier for your reader to understand what you included in your discussion (why you discuss what you did).
One of my professors said, you want to take your reader by the hand from the first sentence. The introduction should lead to the need for the study, the methods how you did the study, the results tell you what you find, and then you discuss your results in relation to the other relevant literature.
Response 5:
Thank you very much for your pertinent comment. On the basis of your comments, we have revised the whole discussion in an attempt to improve it. We think it is now more appropriate. If the reviewer feels that any further changes are necessary, please let us know.
Comment 6:
Language
I think it would be good to remove the ageist and overly judgemental stereotyping language. Stereotyping refers to making generalizations about a group of people and applying them to every individual within that group. This can lead to negative judgments and discrimination. Prejudice is a preconceived opinion, usually negative, that is not based on reason or actual experience.
Original:
When we talk about an ageing population, we think of people who are sicker, with associated comorbidities, dependent and with a gradual loss of functional capacity, and therefore with greater health care needs.
Could be Revised:
When discussing an aging population, it’s important to recognize the diversity of health experiences. While some individuals may experience health challenges, including comorbidities or changes in functional abilities, aging does not inherently equate to dependency or uniform decline. Healthcare needs can vary widely and are influenced by individual circumstances, accessibility to preventative care, and societal support systems.
Response 6:
Thank you very much for your comment. We've reworded the sentence. It's now lines 61-65.
Comment 7:
Original
"The quality of home care is not achieved, since the care provided by IC does not meet the real needs of the dependent person due to the lack of relevant knowledge about health, illness, drug therapy, among others[25]. These difficulties are in line with what has been reported in previous studies, as the training of ICs is crucial if quality care is to be provided [27,28], which can increase the quality of life of ICs that is often diminished by the patient's clinical circumstances, and the quality of life of the patient themselves [29]. The authors confirm the need to study the difficulties experienced by IC, their causes and the impact of IC training on the care they provide [23]. They state that this is a challenge for the population in general and health professionals in particular. Good communication about care needs between the RNS and the caregiver of the older person makes the IC capable to provide the necessary support [24]. One study [20] emphasised that ICs who received practical training through the RNS on techniques such as transferring and positioning the older adult showed significant improvements in the quality of care, resulting in greater safety and comfort for the person.
For the authors, practical training is essential to empower caregivers and reduce the risk of injury. It is also important to emphasise that the domestic reality can be a challenge, as many homes are not prepared to offer the necessary safety to the older adult, which limits the effectiveness of the techniques taught by the RNS [20,23] .
Could be revised:
"Providing quality home care for dependent individuals can be challenging for informal caregivers (ICs), who often take on this role with immense dedication but may lack access to specialized training in areas like health management, illness care, or safe drug administration [25]. Without adequate support, caregivers may face difficulties aligning their care with the evolving needs of their loved ones. Research emphasizes that empowering ICs with tailored education and training—such as guidance on safe patient transfer techniques, home safety adaptations, or stress management—can enhance both the caregiver’s confidence and the care recipient’s quality of life [27–29].
Collaboration between healthcare professionals (e.g., RNS teams) and caregivers is critical. For example, studies show that practical, hands-on training from RNS professionals significantly improves caregivers’ ability to provide safe, comfortable care while reducing risks like injury [20,23]. However, challenges persist, such as home environments that may not easily accommodate safety modifications, highlighting the need for creative, individualized solutions [20,23].
Ultimately, supporting ICs through accessible training, open communication with healthcare teams, and systemic recognition of their invaluable role can help them thrive in their caregiving journey while ensuring the best outcomes for those they care for [24]."
Response 7:
Thank you very much for your comment. We've revised the whole discussion in an attempt to improve it. We think it's more appropriate now. If the reviewer feels that any further changes are necessary, please let us know.
Best regards
Reviewer 4 Report
Comments and Suggestions for Authors
The article addresses a pertinent and current theme, centered on the training of informal caregivers of the elderly by nurses specializing in rehabilitation. The scoping review methodology, anchored in JBI and PRISMA-ScR, is appropriate to the objective and rigorously applied. The title is clear, descriptive and informative. The abstract follows an appropriate structure, however it is suggested to review the expression "value-added support" (line 31), which is vague.
The introduction presents a good contextualization of the problem, with relevant national and international statistics. The Justification is clear, it refers to the importance of training caregivers and the role of specialist nurses. However, some sentences are redundant or too long. A linguistic revision is recommended for greater fluidity and objectivity.
As for the methodology, it presents the appropriate use of the JBI methodology and PRISMA checklist. It clearly describes the inclusion criteria (PCC), research strategy, study selection, and data extraction. It is suggested to better justify the time limitation (last 5 years). To explain, more succinctly, how reliability was ensured among the six reviewers.
As for the results, the seven studies included are clear, with data organized by intervention categories, through detailed presentation in tables. However, it is suggested to reduce the duplication of information between the text and the tables. As well as further clarifying the criteria used to categorize interventions (e.g., "technology" or "emotional support").
As for the discussion, a reflective analysis is evidenced, well anchored in the included literature and in additional studies. It highlights practical implications and relevant recommendations. However, repetition of ideas already expressed in the results section should be avoided. It is suggested to reinforce the originality of the findings identified by this review in comparison with previous reviews.
The conclusion is consistent with the objectives and results, but could be more concise and include explicit recommendations for researchers and policy makers. References are up-to-date and relevant, including national and international sources.
Uniformity in the presentation of DOIs is suggested. Some authors appear with the first initial duplicated or inconsistent abbreviations (e.g., Gomes J., Soares S). The text is understandable, but it would benefit from a native linguistic revision and to improve sentence constructions and avoid repetition.
Comments on the Quality of English Language
The text is understandable, but it would benefit from a native linguistic revision and to improve sentence constructions and avoid repetition.
Author Response
Dear reviewer 4,
Thank you very much for your very helpful comments. We will respond to them and we think that the manuscript has been improved by your suggestions. We will respond in order to improve the manuscript. Please let us know if any further changes are needed.
Comment 1: The article addresses a pertinent and current theme, centered on the training of informal caregivers of the elderly by nurses specializing in rehabilitation. The scoping review methodology, anchored in JBI and PRISMA-ScR, is appropriate to the objective and rigorously applied. The title is clear, descriptive and informative. The abstract follows an appropriate structure, however it is suggested to review the expression "value-added support" (line 31), which is vague.
Response 1:
Thank you very much for your comment. We've deleted the expression ‘support groups’ in line 31, as it doesn't add much to the summary.
Comment 2: The introduction presents a good contextualization of the problem, with relevant national and international statistics. The Justification is clear, it refers to the importance of training caregivers and the role of specialist nurses. However, some sentences are redundant or too long. A linguistic revision is recommended for greater fluidity and objectivity.
Response 2:
Thank you very much for your comment. We've revised the introduction and improved the framing of the issue. We've added lines 43-55; lines 74-88; 100-104.
Comment 3: As for the methodology, it presents the appropriate use of the JBI methodology and PRISMA checklist. It clearly describes the inclusion criteria (PCC), research strategy, study selection, and data extraction. It is suggested to better justify the time limitation (last 5 years). To explain, more succinctly, how reliability was ensured among the six reviewers.
Response 3:
Thank you very much for your comment. We've added lines 154-155; 186.
Comment 4: As for the results, the seven studies included are clear, with data organized by intervention categories, through detailed presentation in tables. However, it is suggested to reduce the duplication of information between the text and the tables. As well as further clarifying the criteria used to categorize interventions (e.g., "technology" or "emotional support").
Response 4:
Thank you very much for your comment. Could you clarify your comment? We've reworked the whole discussion to improve it, and we think it may address your concerns. If not, please let us know. Thank you.
Comment 5: As for the discussion, a reflective analysis is evidenced, well anchored in the included literature and in additional studies. It highlights practical implications and relevant recommendations. However, repetition of ideas already expressed in the results section should be avoided. It is suggested to reinforce the originality of the findings identified by this review in comparison with previous reviews.
Response 5:
Thank you very much for your pertinent comment. On the basis of your comments, we have revised the whole discussion in an attempt to improve it. We think it is now more appropriate. If the reviewer feels that any further changes are necessary, please let us know.
Comment 6: The conclusion is consistent with the objectives and results, but could be more concise and include explicit recommendations for researchers and policy makers. References are up-to-date and relevant, including national and international sources.
Response 6:
Thank you very much for your comment. We've added lines 436-442.
Best regards.
Round 2
Reviewer 2 Report
Comments and Suggestions for Authors
I want to commend the authors for taking time to attend to the comments and suggestions. The manuscript now read very well. I recommend it for publication.
Reviewer 3 Report
Comments and Suggestions for Authors
Thank you. You certainly addressed my concerns. The findings are now well reported and your discussion is excellent. You engage me as a reader with your results and in a discussion about how your results are similar or different than the literature. Well done.